# Yam Staking Reduces Soil Loss Due to Crop Harvesting under Agronomic Management System: Environmental Effect of Soil Carbon Loss

Suarau Oshunsanya [1], Hanqing Yu [1,*], Chibuzo Onunka [2], Victor Samson [3], Ayodeji Odebode [2], Shamsideen Sebiotimo [2] and Tingting Xue [1]

1   Agricultural Clean Watershed Research Group, Institute of Environment and Sustainable Development in Agriculture, Chinese Academy of Agricultural Sciences (CAAS), Haidian District, Beijing 100081, China
2   Department of Agronomy, University of Ibadan, Ibadan 200005, Nigeria
3   Department of Soil Science, Federal University of Agriculture, Makurdi P.M.B. 2373, Nigeria
*   Correspondence: yuhanqing@caas.cn; Tel./Fax: +86-10-82106016

**Abstract:** The staking (elevating creeping vines above the ground with poles) of yam is practiced to optimize crop yield, but its effect on soil loss due to crop harvesting (SLCH-soil adhering to harvested tubers) and its associated carbon loss has not been investigated globally. A 3-year field experiment was conducted to study the yam (*Dioscorea rotundata*) staking effect on SLCH and to examine the environmental effect of soil carbon loss. Staking reduced soil loss due to crop harvesting by 55.6% and increased yam yield by 33.3% when compared to un-staking. Soil carbon loss and root hair weight per tuber yield decreased by 47.7 and 58.4%, respectively, under staking compared with un-staking practices. The un-staking with higher moisture ($\simeq$42%) exported two times more soil-available nutrients (N, P, K and Ca) than staking. SLCH was also linearly related to root hair weight ($R^2 = 0.88$–0.75; $p < 0.05$) and moisture content ($R^2 = 0.79$–0.63; $p < 0.05$). The lower ratio of root hair weight to tuber yield coupled with moderate mound moisture in staking neutralized its higher tuber yield effect on SLCH by reducing soil loss and its carbon loss. Thus, yam staking mitigates soil loss and its carbon loss which can increase the sequestration potential of soil carbon stock.

**Keywords:** soil loss; carbon loss; yam staking; soil moisture; root hair weight; agronomic practice; tuber yield



## 1. Introduction

Agronomic practices are agricultural operations that farmers incorporate into their farm management systems to improve soil environmental quality, enhance water use efficiency, manage crop residues, reduce soil erosion and ultimately increase crop yields [1]. With good agronomic practices, 18–20 t ha$^{-1}$ of yams can be obtained from the field [1–3]. Twining the vines of yam around the pole (staking) or leaving it on the surface soil (un-staking) is one of the traditional agronomic practices employed for cultivation [4].

Staking is defined as the act of elevating creeping vines above ground level using supporting structures for optimum crop yield. Staking has been reported to influence the growth and yield of some varieties of potato, cowpea, yam and vegetables [5]. This is because the staking of crops allows for maximum sunlight interception with leaves for photosynthesis [6]. It protects crop plant vines from physical damage as well as reducing the spread of soil-borne diseases from attacking the growing vines [7,8]. Staking enhances opportunities for effective control of pests and weeds [9]. In addition, the staking of crop vines assist, to a large extent, in the control of anthracnose disease, which is usually common during the rainy season [8]. Staking is usually done before the sprouting of crops or when the vines are long enough to be twisted around the stakes [10]. The pyramid method of staking (arranging three or four stakes to form a pyramid structure) is most commonly used



by farmers in many countries [9] for the following reasons: (i) It suppresses weeds for better crop performance as well as contributing to easy uplifting of roots and tubers during crop harvesting. (ii) It also provides the optimal soil moisture and environmental temperature for growing yam, which is one of the most prominent root and tuber crops grown all over the world to provide food for over two billion people [11] and has both agricultural and cultural importance, especially in Asia, Africa, Oceania and Latin America [12]. With the aforementioned influence of staking on the mound environment, there is a possibility or tendency that staking may likely affect soil loss due to crop harvesting.

The harvesting of root, tuber and bulb crops have been documented to cause a substantial loss of soil from the field [13]. Over the years, potato, cocoyam, yam, ginger, turmeric, sugar beet, chicory, carrot and onion have been widely assessed for their potentials for soil loss due to crop harvesting (SLCH) in many countries [13–27]. It contributes to about 50% of the decline in soil productivity of agricultural lands [27,28]. More importantly, soil loss due to crop harvesting can contribute to global warming by reducing the sequestration potential of soil carbon stock. For instance, [29] reported that the world's surface soil (where root and tuber crops are harvested) is a carbon sink for approximately twice as much greenhouse gas as found in the entire atmosphere. Thus, the removal of top soil during harvesting will lead to more greenhouse gasses and consequently an increase in global warming. Despite the negative impact of soil loss due to crop harvesting on soil productivity and the environment, it is still being given little or no attention globally. Nevertheless, the contribution of SLCH to global soil loss and environmental degradation cannot be ignored.

Many scholars have studied the key factors affecting SLCH. Ruysschaert et al. [17] identified four groups of factors (soil, crop morphology, harvest technique and agronomic practices) that significantly determined SLCH. Among the aforementioned factors, it is only the impact of staking on SLCH that has not been investigated globally. For instance, several research studies [17,22,30] have been conducted on the impact of soil properties (moisture content, soil texture, soil structure and soil organic matter) on SLCH. Another group of researchers [17,22,25,26,30] investigated the impact of crop morphology (root hair abundance, root shape, root cortex, rootlets, spherical and elongated roots) on SLCH. In the same way, [31,32] conducted experiments on the influence of harvest techniques (harvesting depth, rate of harvesting and shape of harvesting machine) on SLCH. However, as an important aspect of yam production, staking influence on SLCH has not been investigated. Therefore, the objectives of this experiment were to study the influence of yam staking on SLCH and its associated carbon loss, and to understand the mechanisms by which staking influence SLCH and carbon loss under an agronomic management system. Such information can be used to assess the level of soil degradation associated with the use of agronomic practices for soil conservation programs and soil carbon sequestration. Understanding the relationship between staking and SLCH can also provide room for crop improvement to help the crop reach its full potential for food security.

## 2. Materials and Methods

### 2.1. Site Description

The experiment was conducted at the Teaching and Research Farm (Latitude 07°27′05.6″ N, Longitude 03°53′31.9″ E) of the University of Ibadan, Nigeria (Figure 1). The site has an altitude of 196 m above sea level with a tropical climate condition. The rainfall pattern is bimodal with an average of 1230 mm per annum. The rainfall amount during the study period (2014–2016) is presented in Figure 2. There are two growing seasons: an early season (March to July) and a late season (late August to early November). The soil was well drained and belonged to an Alfisol formed on a granite gneiss parent material. The site was established on a land made up of traditional mounds which had been fallowed for 5 years before the commencement of this study. Fertilizer NPK 15-15-15 was only applied during the third growing season. Soil texture (0–30 cm) at the commencement of the experiment was loamy sand with a bulk density of 1.33 Mg m$^{-3}$, pH of 6.7 and organic carbon of 29.2 g kg$^{-1}$. Available nitrogen, phosphorus and potassium were 3.4 g kg$^{-1}$, 38 mg kg$^{-1}$ and 0.4 cmol kg$^{-1}$, respectively.

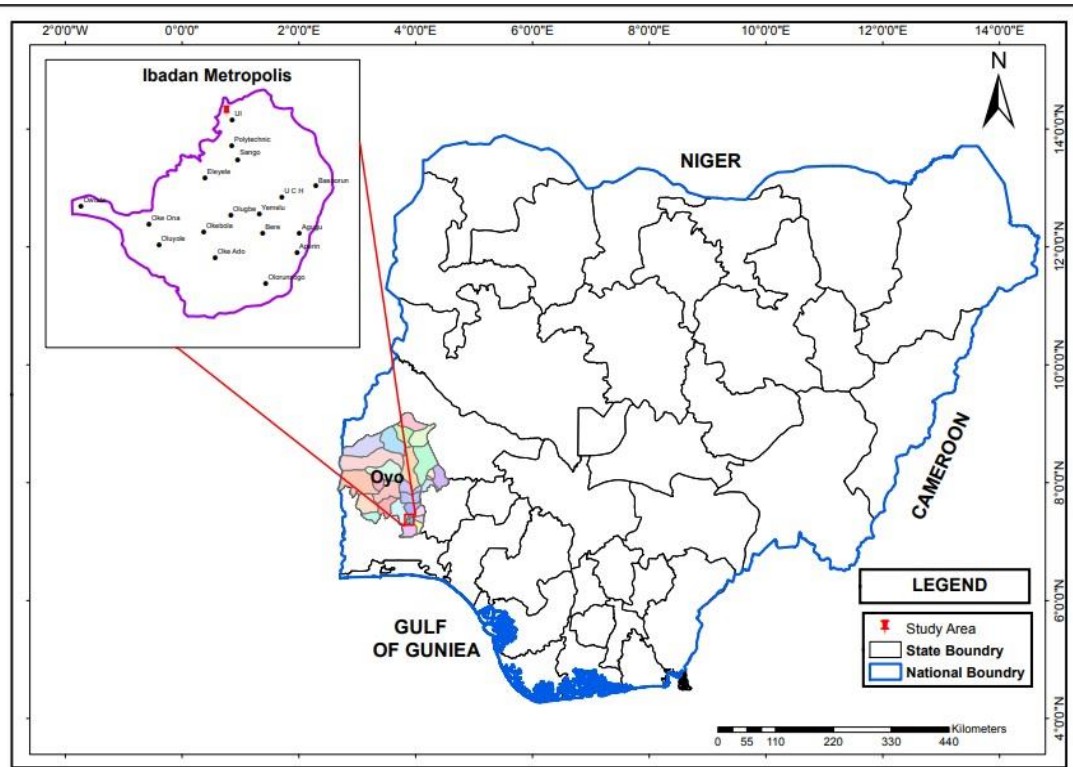

**Figure 1.** A map showing the location of the study site.

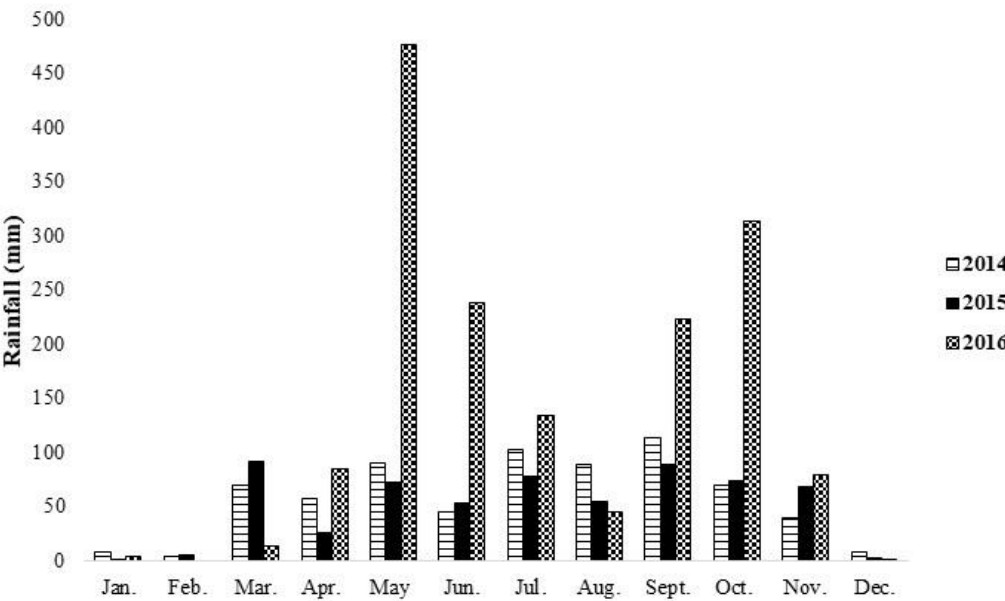

**Figure 2.** Rainfall amount for 2014, 2015 and 2016 during the study period.

### 2.2. Soil Sampling and Analysis

Initial soil samples ($n$ = 12) were collected at the commencement of the experiment on 25 February 2014 to ascertain the baseline properties. Subsequent samples ($n$ = 12 × 3 = 36) were collected at every harvest (12 November 2014; 20 November 2015; 14 November 2016) to monitor changes in soil status at harvest. All samples were taken at the center of the mound, 0–30 cm depth, where yam tubers were harvested. The soil auger was used to collect samples for chemical analysis while the cylindrical core (5 cm height by 5 cm diameter) was used for physical properties as described by [33]. Samples for chemical

properties were sieved (2 mm for P, K, Ca, and Mg and 0.5 mm sieves for N and C) and stored in the soil room ($-4$ °C). Samples for particle size analysis were prepared with a 2 mm sieve. The soil pH was measured in a medium which contained soil and distilled water with a 1:1 ratio [33] using a pH meter. The soil organic carbon content was determined using the Walkley–Black wet-oxidation method [34] as described by [35], while the available nitrogen content was determined using the macro-Kjeldahl digestion-distillation apparatus following the [36] procedure. Melich III was used to extract the plant-available phosphorus and the absorbance was read using a spectrophotometer [37]. Ca and Mg were determined using neutral ammonium acetate as the extractant [38]. Subsequently, the extract using the 0.01 M EDTA (ethylene di-amine tetra-acetic acid) titration method was used to obtain Ca and Mg, while potassium was determined using a flame photometer. The particle size distribution of the soil was determined using the Bouyoucos hydrometer method as described by [39]. After oven-drying the soil samples to constant weight at 105 °C, the soil bulk density was ($\rho$) computed using Equation (1) as described by [39]. Gravimetric moisture content and volumetric moisture content (Vm) were also computed using Equations (2) and (3) as described by [39].

$$\rho = \frac{Ms}{Vb} \tag{1}$$

where $\rho$ is the bulk density (Mg m$^{-3}$), $Ms$ is the mass of oven-dry soil (Mg) and $V_b$ is the soil bulk volume (m$^3$).

$$Total\ porosity = \frac{(saturated\ mass\ of\ soil) - (oven\ dry\ mass\ of\ soil)}{soil\ volume} \tag{2}$$

Total porosity was calculated as presented in Equation (2):

Volumetric moisture content ($\theta$) was computed by multiplying gravimetric moisture content ($\rho m$) by bulk density ($\rho$) as shown in Equation (3):

$$\theta = \rho m \times \rho \tag{3}$$

### 2.3. Planting and Staking of Yam Vines

The experimental plot was manually cleared with a machete at the commencement of the study. Staking and un-staking treatments were randomly assigned and replicated six times. Thirty traditional mounds each occupied a unit plot of 10 m × 3 m. Mounds were spaced at 1 m × 1 m. Traditional mounds were constructed using a traditional hoe. Mounds were made by using a hoe to scrap topsoil together at the center of a 1 m$^2$ land area to form a cone shape. Mounds were constructed to the same perpendicular height (72 cm) and circumference (270 cm) at the beginning of the experiment. A total of 360 mounds were made for all experimental plots on the field. A clean knife dipped in benomyl (benlate at 2 tsp/4 L of water) fungicide was used to cut the yam tuber into yam setts of 250 g weight and subsequently dipped into benlate for at least 10 s and allowed to dry before planting. The yam setts were then planted directly at the center top of the mounds to a depth of 15 cm. The impact of staking and un-staking of yam vines on mound dimensions (slant height, perpendicular height, base circumference and volume) were assessed at the end of every growing season to determine mound size effect on moisture content. In staking, the pyramid method was adopted for this study because the method is commonly practiced in many countries [40]. This method helps farmers to reduce weed growth in the field. In this study, stakes of about 3 m long were erected beside the mounds in a slant form such that the heads of a group of four to six stakes converged into a pyramid shape over the mounds (Figure 3). Yam vines were staked at 3 weeks after sprouting when vines were long enough to be twisted around the stake without breakage. When vines were fully twisted to the top of the pyramid, vines interwoven around the pyramid could hardly be penetrated by sunlight and rainwater. This arrangement tends to expose foliage to direct sun rays and the provided shade conserves soil moisture by reducing evaporation from the surface soil.

Weeding along furrows and around the field was done with the aid of a hoe at 3-month intervals. However, the hand picking of weeds was adopted on the surface and on the base of the mounds to minimize soil disturbance.

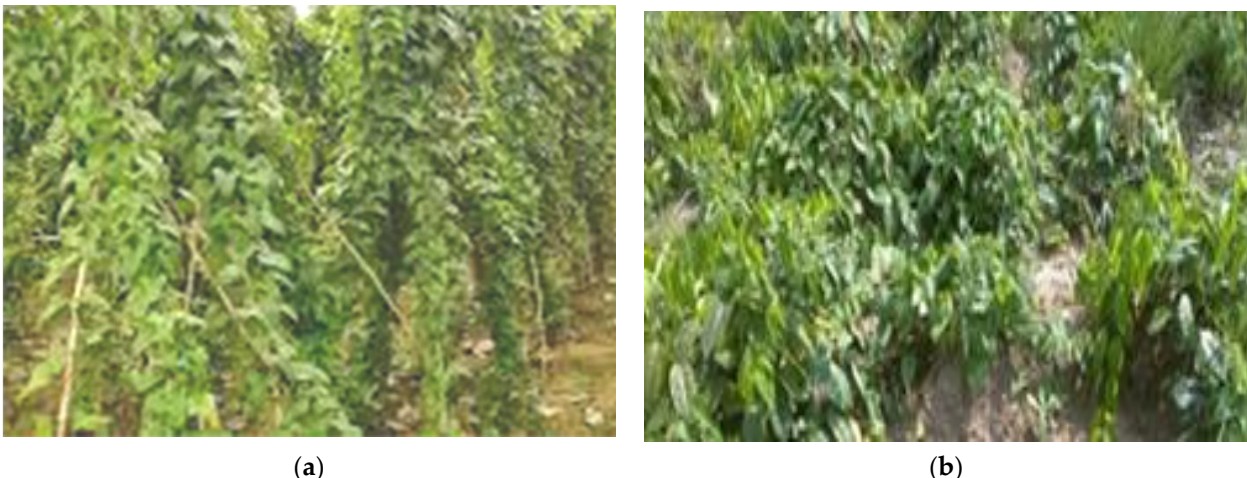

| (**a**) | (**b**) |

**Figure 3.** Yam (*Dioscorea rotundata*) farming under staking and un-staking of vines in traditional agriculture: (**a**) staked (pyramid method) yam vines; (**b**) un-staked yam vines.

### 2.4. Estimation of above Ground Total Biomass, Tuber Yield, and Soil and Nutrient Losses

Above-ground biomass was harvested at ground level and weighed separately. Yam tubers were harvested manually by removing matured yam tubers with a machete. Harvesting was carried out in November 2014, 2015 and 2016. Soil adhering to yam tubers was removed by shaking [20] on the field. Subsequently, partially clean yam tubers were washed to remove the fine soil particles adhering to the tubers in the laboratory. Soils obtained by shaking and washing of tubers were combined to determine soil loss on a plot-to-plot basis. Root hairs on the tubers were carefully removed with a sharp knife and weighed. Above-ground biomass dry matter was determined by rinsing the yam shoot (leaves and stems) with deionized water before oven drying at 70 °C to a constant weight.

Yam tuber yield per land area per harvest (*Nty*) was estimated using Equation (4) as described by [24]. Wet soil loss was dried and weighed before analyzing for nutrient loss using standard procedures as described under soil analysis. Soil loss by unit crop yield (*SLCHspec*) and soil loss due to crop harvesting (*SLCHcrop*) was estimated using Equations (5) and (6), respectively, as described by [17]. Nutrient loss (expressed on an elemental basis) was estimated using Equation (7) as described by [17]. *Mds* is the mass of dry soil (kg), *Mrf* is the mass of rock fragments which is zero, and *Mcrop* is the mass of yam tubers (kg).

$$Nty \left( kg\ ha^{-1}\ harvest^{-1} \right) = Tuber\ yield\ (kg) / Land\ area\ (ha) \tag{4}$$

$$SLCHspec \left( kg\ kg^{-1} \right) = (Mds + Mrf) / (Mcrop) \tag{5}$$

$$SLCHcrop \left( kg\ ha^{-1}\ harvest^{-1} \right) = SLCHspec \times Nty \tag{6}$$

$$Nutrient\ loss \left( kg\ ha^{-1}\ harvest^{-1} \right) = \frac{Nutrient\ (g)}{Soil\ (100\ g)} \times 0.01 \times SLCHcrop \tag{7}$$

### 2.5. Data Analysis

Data collected were subjected to a normality test (Leven's test) to verify the distribution pattern of the variables before statistical analysis. Data were normally distributed for the 3 years. Data on tuber yield, root hair weight, soil loss, soil physical properties and soil

nutrient loss for years 1–3 were subjected to T–test analysis at $\alpha = 0.05$ by using Genstat 5 release 3.2 (PC/Window 95). Origin pro 2021 was used to compare the contribution of staking and un-staking to soil loss due to yam harvesting using a violin plot. Pearson's correlation coefficients and significance levels were determined between dependent variables (soil loss due to crop harvesting and soil loss by unit crop yield) and independent variables (yam tuber yield, root hair weight, moisture content, bulk density, sand, silt, clay and carbon).

## 3. Results

### 3.1. Soil and Nutrient Losses Due to Crop Harvesting

Soil loss due to yam harvesting under staking and un-staking are presented in Figure 4. Yam staking significantly ($p < 0.05$) reduced soil loss due to crop harvesting during the study period.

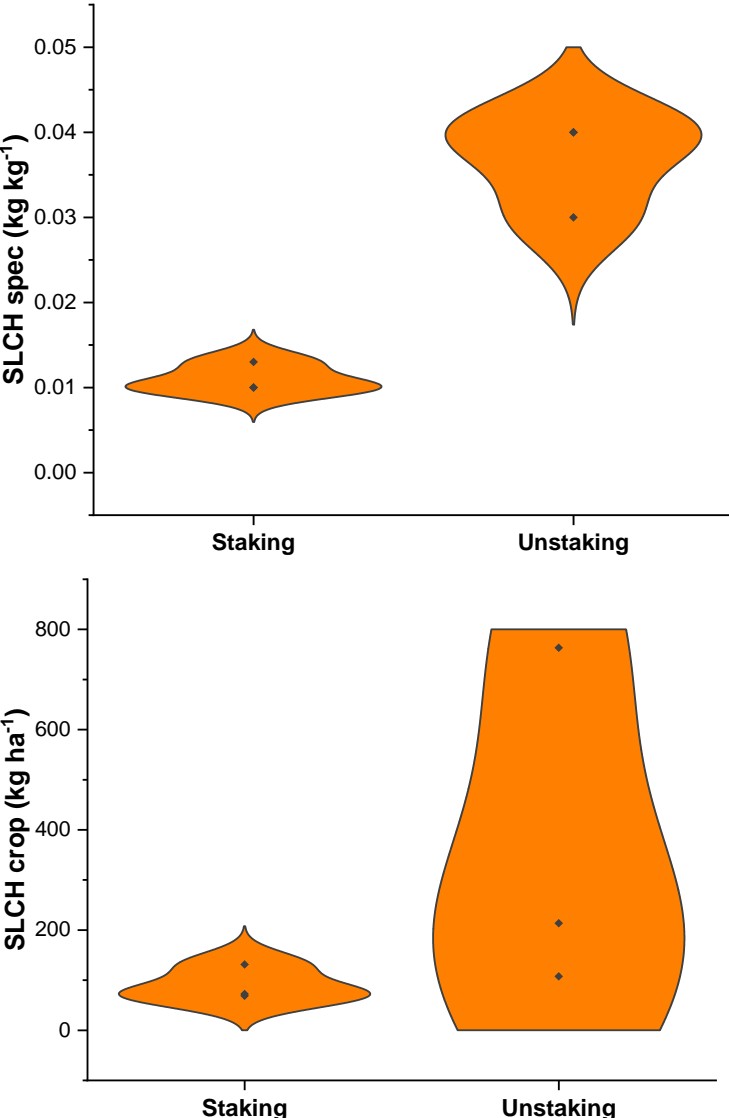

**Figure 4.** Quantitative comparison of soil loss due to the harvesting of yam under staking and unstaking in a traditional farming system. Orange balloons represent the quantity of soil loss, while black dots represent the concentration/number of soil samples. SLCHspec and SLCHcrop represent soil loss due to unit yield crop harvest and soil loss due to crop harvest, respectively.

Soil loss by unit crop yield (SLCHspec) was, on average, higher under un-staking compared to staking by 72.2.0% over the 3 years (Figure 4). As expected, soil loss due to crop harvesting (SLCHcrop) exhibited a similar pattern to SLCHspec. SLCHcrop was greater under un-staking compared to staking by 55.6% (Figure 4). The amount of available lost with crop harvest is a factor determining the quality of SLCH. In this present study, soil loss under un-staking contained 36–46% more soil available nutrients (C, N, P, K, and Ca) than staking (Figure 5). On average, staking reduced available nutrient loss by 42% compared with un-staking practice. More importantly, soil carbon loss, which can influence the sequestration potential of soil carbon stock, was reduced under staking compared to un-staking for years 1–3 by 50.0%, 44.0% and 42.9%, respectively, with a mean of 45.6%.

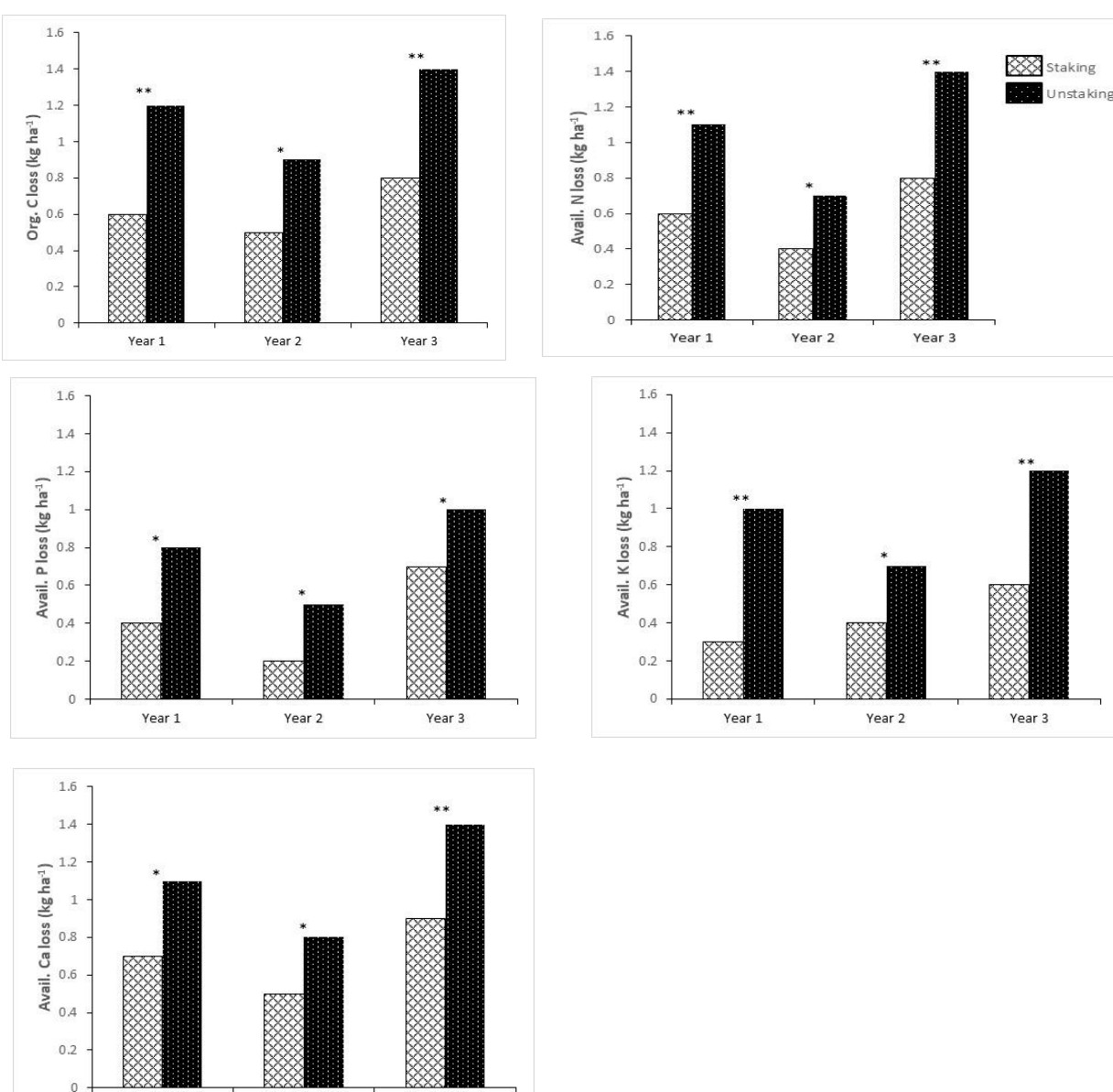

**Figure 5.** Influence of staking and un-staking practices on soil nutrient losses due to yam harvesting under a traditional farming system. * $p < 0.05$ and ** $p < 0.01$ indicate significant difference between staking and un-staking at 5 and 10% probability levels, respectively, for each year. Years 1, 2 and 3 = yam growing cycles in 2014 ($n = 24$), 2015 ($n = 24$) and 2016 ($n = 24$), respectively.

### 3.2. Above-Ground Biomass, Root Hair Weight and Tuber Yield

Staking affected the growth and development of root hairs. Root hair weight was 36.4% higher in un-staking than staking (Table 1). Staking practice did affect the number of root hairs per tuber yield. Root hair weight per tuber yield was 58.4% higher in un-staking compared to staking. Nevertheless, staking strongly influenced above-ground biomass and tuber yield (Table 1). Staking increased above-ground biomass dry matter compared to un-staking by 31.1%. Tuber yield was strongly affected by above-ground biomass. Proportionately, above-ground biomass increased tuber yield by a factor of 4.7 for staking and a factor of 4.4 for un-staking. Overall, above-ground biomass in staking explained considerably more ($\geq$11%) of the tuber yield than those of un-staking. Tuber yield was higher in staking than un-staking by an average of 34.3% over 3 years (Table 1).

**Table 1.** Influence of yam staking on above-ground biomass dry matter, root hair weight and tuber yield under a traditional farming system.

| Parameter | Staking | Un-Staking | *T*-Test (*p*-Value) | Staking | Un-Staking | *T*-Test (*p*-Value) | Staking | Un-Staking | *T*-Test (*p*-Value) |
|---|---|---|---|---|---|---|---|---|---|
| | | Year 1 | | | Year 2 | | | Year 2 | |
| AGBDM (kg ha$^{-1}$) | 1624.61 | 1225.37 | 2.45 (0.005) | 1462.18 | 831.42 | 21.4 (0.001) | 2765.45 | 2067.11 | 19.32 (0.001) |
| RHW (kg ha$^{-1}$) | 8.82 | 14.10 | 1.37 (0.001) | 6.40 | 9.87 | 1.39 (0.001) | 29.70 | 32.60 | 8.16 (0.001) |
| Tuber yield (t ha$^{-1}$) | 7.22 | 5.34 | 1.09 (0.001) | 6.86 | 3.58 | 1.05 (0.001) | 13.11 | 9.27 | 1.79 (0.001) |
| RHW/Tuber yield | $1.22 \times 10^{-3}$ | $2.64 \times 10^{-3}$ | 0.001 (0.01) | $9.33 \times 10^{-4}$ | $2.76 \times 10^{-3}$ | 0.001 (0.01) | $2.27 \times 10^{-3}$ | $3.52 \times 10^{-3}$ | 0.001 (0.05) |

AGBDM = Above-ground biomass dry matter; RHW = Root hair weight; Years 1, 2 and 3 = Yam growing cycles in 2014 (*n* = 24), 2015 (*n* = 24) and 2016 (*n* = 24), respectively.

### 3.3. Relationships of SLCH with Soil Physical Indices and Root Crop Morphology

Table 2 presents the impact of staking on soil physical properties at harvest. Particle size distribution (coarse sand, fine sand, silt and clay) and soil bulk density were not affected by staking. However, mound moisture was significantly (*p* < 0.05) influenced by staking, such that the mound moisture content was 16.5% higher in un-staking compared to staking. Table 3 presents the characteristic effects of soil carbon, soil physical properties, root hair weight, tuber yield and root hair weight per tuber yield on sediment exportation. Root morphological characteristics strongly correlated ($r^2 \simeq 0.81$) to soil loss under staking and un-staking practices. Among soil physical properties, soil moisture was distinctly related ($r^2 \simeq 0.77$) to soil loss. However, other soil physical properties (sand, silt, bulk density etc.) were poorly related to soil loss. The major soil loss was related to root morphologies (root hair weight per tuber yield) and mound moisture at harvesting time.

**Table 2.** Influence of yam staking on soil physical properties during harvesting periods of 2014, 2015 and 2016 under a traditional farming system. ns = not significant. ears 1, 2 and 3 represent the time of harvesting in November of 2014, 2015 and 2016, respectively. LS = Loamy sand; $\theta$ = volumetric moisture content.

| Parameter | Staking | Un-Staking | *T*-Test (*p*-Value) | Staking | Un-Staking | *T*-Test (*p*-Value) | Staking | Un-Staking | *T*-Test (*p*-Value) |
|---|---|---|---|---|---|---|---|---|---|
| | | Year 1 | | | Year 2 | | | Year 3 | |
| Coarse sand (g kg$^{-1}$) | 625.4 | 623.4 | ns (ns) | 631.0 | 633.0 | ns (ns) | 628.0 | 629.0 | ns (ns) |
| Fine sand (g kg$^{-1}$) | 216.0 | 218.0 | ns (ns) | 211.2 | 207.7 | ns (ns) | 211.2 | 209.6 | ns (ns) |
| Silt (g kg$^{-1}$) | 63.0 | 63.0 | ns (ns) | 60.8 | 62.6 | ns (ns) | 61.8 | 63.5 | ns (ns) |

**Table 2.** *Cont.*

| Parameter | Staking | Un-Staking | *T*-Test (*p*-Value) | Staking | Un-Staking | *T*-Test (*p*-Value) | Staking | Un-Staking | *T*-Test (*p*-Value) |
|---|---|---|---|---|---|---|---|---|---|
| | | Year 1 | | | Year 2 | | | Year 3 | |
| Clay (g kg$^{-1}$) | 95.6 | 95.6 | ns (ns) | 97.0 | 96.7 | ns (ns) | 99.0 | 97.9 | ns (ns) |
| Textural class | LS | LS | | LS | LS | | LS | LS | |
| Bulk density (Mg m$^{-3}$) | 1.38 | 1.39 | ns (ns) | 1.42 | 1.41 | ns (ns) | 1.45 | 1.45 | ns (ns) |
| $\theta$ (m$^3$ m$^{-3}$) | 0.19 | 0.32 | 0.02 (0.01) | 0.14 | 0.27 | 0.02 (0.01) | 0.18 | 0.36 | 0.01 (0.001) |

**Table 3.** Correlation matrix between SLCH and root hair density, tuber yield, root hair density per tuber yield, sand, silt, clay, bulk density and moisture under traditional agriculture.

| | SLCHcrop | TY | RHW | Sand | Silt | Clay | BD | Moisture | RHW/TY | AGMDM |
|---|---|---|---|---|---|---|---|---|---|---|
| | **Staking** | | | | | | | | | |
| SLCHspec | 0.94 *** | 0.58 * | 0.96 *** | −0.10 | 0.12 | 0.09 | −0.42 * | 0.92 *** | 0.65 * | 0.46 * |
| SLCHcrop | | 0.55 * | 0.91 *** | −0.17 | 0.08 | 0.12 | −0.11 | 0.84 *** | 0.61 * | 0.43 * |
| TY | | | 0.91 *** | −0.30 | 0.02 | 0.29 | −0.55 * | 0.54 * | 0.64 * | 0.74 ** |
| RHW | | | | 0.11 | −0.42 * | 0.15 | −0.47 * | 0.88 *** | 0.89 *** | 0.88 *** |
| Sand | | | | | −0.38 * | −0.72 ** | −0.05 | −0.25 | −0.01 | −0.21 |
| Silt | | | | | | −0.36 | 0.01 | 0.10 | 0.12 | 0.37 |
| Clay | | | | | | | 0.05 | 0.18 | −0.06 | 0.42 |
| BD | | | | | | | | −0.05 | −0.45 * | −0.11 |
| Moisture | | | | | | | | | 0.76 ** | 0.74 ** |
| RHW/TY | | | | | | | | | | 0.58 * |
| | **Un-staking** | | | | | | | | | |
| SLCHspec | 0.95 *** | 0.76 ** | 0.96 *** | 0.09 | 0.02 | 0.11 | −0.16 | 0.70 ** | 0.59 * | 0.42 * |
| SLCHcrop | | 0.86 *** | 0.88 *** | −0.14 | 0.09 | 0.10 | −0.12 | 0.89 *** | 0.58 * | 0.41 * |
| TY | | | 0.55 * | −0.23 | 0.11 | 0.19 | −0.28 | 0.75 ** | 0.60 * | 0.63 * |
| RHW | | | | −0.10 | −0.12 | 0.19 | −0.27 | 0.78 ** | 0.81 ** | 0.75 ** |
| Sand | | | | | −0.48 * | −0.79 ** | −0.20 | −0.05 | −0.22 | −0.05 |
| Silt | | | | | | −0.15 | 0.27 | 0.03 | 0.02 | 0.01 |
| Clay | | | | | | | 0.04 | 0.04 | 0.22 | 0.23 |
| BD | | | | | | | | −0.03 | 0.47 * | −0.39 |
| Moisture | | | | | | | | | 0.71 ** | 0.52 * |
| RHW/TY | | | | | | | | | | 0.48 * |

*** = $p < 0.001$, ** = $p < 0.01$, * = $p < 0.05$. TY = tuber yield, RHW = root hair weight, BD = bulk density, AGMDM = above-ground biomass dry matter. The data used for correlation analysis were the 3-year (2014, 2015 and 2016) data (*n* = 36) pooled together separately under staking and un-staking.

## 4. Discussion

### 4.1. Effects of Mound Moisture and Root Hairs on SLCH

Mound moisture content and root hair weight per yam tuber affected the sediment adherence and exportation differently under staking and un-staking practices. Mound moisture was about 16% in staking compared to un-staking. Staking optimizes mound moisture content due to the pyramid configuration arrangement (staking structure) which allows cross ventilation with little sunlight penetration (Figure 3). Moisture content enhances the sediment adherence to the surface crop [22,26]. Thus, optimization of mound moisture content by staking practices can decrease SLCH. This is supported by the linear positive relationship between mound moisture content and SLCHspec (r = 0.70–0.92; $p < 0.001$) (Table 3). A similar report made by [18] suggested that differences of 74 to 79%

in SLCH can be due to variation in soil water and clay content. Many researchers have reported soil moisture content as one of the most important factors controlling soil loss due to crop harvesting [17,19,22,32,41–47]. For instance, soil moisture content was found to be a significant ($p < 0.05$) variable that accounted for variations in soil loss due to crop harvesting [22,47]. In another experiment conducted in Turkey, [32] discovered that clay, lime, organic matter and soil moisture content accounted for 35% of the variation in soil loss due to crop harvesting. Thus, agronomic practices that support the reduction in sediment adherence and exportation which can make mounds dry when harvesting should be considered by farmers. A promising practice is staking, which has the potential to optimize moisture content at the point of harvesting without posing a negative impact on crop yield. On the other hand, the overlapping of crawling vine leaves on mound surface prevented air circulation, resulting in higher mound moisture in un-staking and consequently increasing SLCH (Figure 3). Indeed, the un-staked mound yam condition (higher moisture content) supported a higher percentage of root hairs per tuber yield compared to staking (Table 1). [48,49] reported that moisture content played a primary role in regulating root hair growth in soils. The literature also reports that root hairs are agents used for trapping sediment and exporting nutrients from the field during crop harvesting [24,25,42,50]. Thus, the lower ratio of root hairs to tuber yield might be responsible for lower SLCH under staking. The root morphologies are important determinants of SLCH. For instance, [26] reported that the elongated potatoes exported appreciably more sediment than the spherical potatoes by a range of 16 to 33% in Brazil. Moreover, soil loss and carbon loss at harvest varied over the 3 years which could be attributed to changes in rainfall amount during the study period. [24] made similar observations that SLCH and soil moisture content depend on rainfall events over the 3 years of study in Nigeria.

*4.2. Influence of Staking on Nutrient Loss*

Staking reduces soil nutrient loss due to yam harvesting under traditional agriculture (Figure 5). Nutrient loss affects land degradation through nutrient exportation due to harvesting. Nutrient adherence in this study was lower in staking compared to un-staking. The literature reports that moisture and root morphologies affect nutrient exportation. Moisture enhances nutrient adherence to tubers during harvesting [25,41,47]. Here, the general nutrient loss pattern was similar to that stated in the literature, that is, the available nutrient losses (N, P, K, Ca and C) were higher in un-staking with a high soil moisture content ($\simeq 42\%$) which exported two times more nutrients than staking with a low moisture content. Higher root hairs per unit yam tuber ($\simeq 58\%$) in un-staking might have been responsible for its available nutrient loss. The un-staked yam morphological characteristics can affect the nutrient adherence at distinct root hairs. In this study, harvesting yams under staking with root hairs per unit tuber yield of approximately 7% reduced the available nutrient loss by a factor of almost two. Farmers should therefore avoid practicing un-staking to mitigate soil nutrient loss due to harvesting. In terms of pollution, nutrient losses can contribute to environmental degradation if not curtailed. For instance, available nitrogen and phosphorus losses due to harvesting can be discharged into the aquatic ecosystem where it can cause eutrophication [25,44]. Moreover, comparing available nutrient loss due to yam harvest with other crops showed that available nutrient losses varied across countries. For instance, [23] reported that the potato harvest in China produced available N ($2.22$ kg ha$^{-1}$ yr$^{-1}$) loss which was higher than the available N ($0.83$ kg ha$^{-1}$ yr$^{-1}$) loss from this study by 62.6%, while available P ($0.6$ kg ha$^{-1}$ yr$^{-1}$) loss from this study was higher than the available P ($0.15$ kg ha$^{-1}$ yr$^{-1}$) loss due to the potato harvest in China by 75.0%. For other countries, available N and K losses in this study were lower than the available N and K losses due to the harvesting of garlic ($11.90$ kg N ha$^{-1}$ yr$^{-1}$ and $4.00$ kg K ha$^{-1}$ yr$^{-1}$), potato ($2.27$ kg N ha$^{-1}$ yr$^{-1}$ and $1.06$ kg K ha$^{-1}$ yr$^{-1}$), sugar beet ($3.35$ kg N ha$^{-1}$ yr$^{-1}$ and $1.74$ kg K ha$^{-1}$ yr$^{-1}$) and radish ($9.90$ kg N ha$^{-1}$ yr$^{-1}$ and $7.45$ kg K ha$^{-1}$ yr$^{-1}$) in Iran [41], and sugar beet ($1.84$ kg N ha$^{-1}$ yr$^{-1}$ and $0.91$ kg K ha$^{-1}$ yr$^{-1}$) in Turkey [43]. However, the P loss from this study was greater than the P loss due to the harvest-

ing of cassava (0.16 kg P ha$^{-1}$ yr$^{-1}$) in Uganda [44]; onion (0.05 kg P ha$^{-1}$ yr$^{-1}$), carrot (0.08 kg P ha$^{-1}$ yr$^{-1}$), potato (0.01 kg P ha$^{-1}$ yr$^{-1}$) in Tanzania (Mwango et al. 2015b); and sugar beet (0.02 kg P ha$^{-1}$ yr$^{-1}$) in Turkey [46].

*4.3. Environmental Implication of Soil Carbon Loss*

Staking reduces soil carbon loss due to yam harvesting under traditional agriculture (Figure 5). Soil loss due to crop harvesting reduced soil potential for carbon sink. This is because soil loss contains an appreciable amount of carbon content which is removed along with the soil during the harvesting of root and tuber crops from the field. Continuous exportation of soil carbon content during harvesting will lower the topsoil (soil removal will cause a decrease change in the soil carbon content) which is a carbon sink for approximately twice as much greenhouse gas as in the entire atmosphere [29]. The environmental implication of the reduction in soil carbon loss by staking supports the sequestration potential of soil carbon stock and consequently reducing global warming. On the other hand, higher soil carbon loss due to crop harvesting under un-staking will contribute to global warming by reducing the sequestration potential of soil carbon stock. Thus, farmers should consider the staking of yams for soil conservation and environmental implications.

*4.4. Possible Effect of above-Ground Biomass on Crop Yield*

The amount of above-ground biomass strongly determines the tuber yield under both staking and un-staking (Table 3). Higher above-ground biomass under staking can be due to the adequate display of leaves on the pyramid frame which provides room for maximum sunlight interception for photosynthesis, thus leading to more photosynthetic products [51,52]. Consequently, more photosynthetic products usually translate to a higher crop yield [53]. The higher tuber yield recorded under staking agreed with earlier findings of [8,54,55], who reported an increase in tuber yield due to the staking of yam. However, tuber yield was significantly ($p < 0.01$) linearly related to soil loss due to crop harvesting (Table 3). The yam yield effect on SLCH under staking was dwarfed by the lower mound moisture coupled with reduced root hairs per tuber yield. Root hair weight explained about 88 to 96% of SLCH, while tuber yield only accounted for 19 to 24% of the variation in SLCH (Table 3). In this study, the ratio of root hairs to tuber yield was a driving force for sediment exportation.

**5. Conclusions**

Staking reduces soil loss and its associated carbon loss during yam harvesting under traditional agriculture due to moderate mound moisture and lower root hairs per unit tuber yield under yam staking practice. Moisture content and root hair weight per tuber yield are the major factors regulating soil loss and its associated carbon loss during yam harvesting. Moisture content and root hairs are the agents of adherence and exportation of soil loss, carbon nutrient loss and other available nutrient (N, P, K and Ca) losses during the harvesting of crops. Reduced soil moisture content coupled with lower root hair weight per tuber yield could have been responsible for lower soil available nutrient losses during harvesting under staking. The effect of higher tuber yield on soil nutrient losses due to crop harvest under staking was neutralized by its lower root hairs per unit tuber yield. However, higher root hair weight per tuber yield and higher moisture content under un-staking contributed to its higher carbon nutrient adherence during harvesting. The results further show that soil loss due to the harvesting of crops can contribute to global warming by reducing the sequestration potential of soil carbon stock. Thus, agronomic practices, such as staking, should be encouraged among farmers for optimum economic yield and mitigation of soil carbon loss due to crop harvesting. Further studies should focus on the impact of soil loss due to crop harvesting on soil carbon sequestration in a broad soil texture and climatic conditions.



**Author Contributions:** Conceptualization, S.O. and H.Y.; methodology, C.O. and S.S.; software, V.S.; validation, S.O., H.Y. and V.S.; formal analysis, C.O.; investigation, S.S.; resources, H.Y.; data curation, S.O.; writing—original draft preparation, S.O., A.O. and T.X.; writing—review and editing, S.O., H.Y., A.O. and T.X.; visualization, S.O.; supervision, H.Y.; project administration, H.Y. and S.O.; funding acquisition, H.Y. All authors have read and agreed to the published version of the manuscript.

**Funding:** This research was funded by the National Key Research and Development Program of China, grant number 2017YFC0505402" and also partly funded by the National Natural Science Foundation of China, grant number 31000944".

**Data Availability Statement:** Not applicable.

**Acknowledgments:** The authors would like to appreciate the support received from both field and laboratory staff of the Department of Agronomy, University of Ibadan, Nigeria. The authors also acknowledge the support of Agricultural Clean Watershed Research Group, Institute of Environment and Sustainable Development in Agriculture, Chinese Academy of Agricultural Sciences (CAAS), Haidian District, Beijing 100081 China.

**Conflicts of Interest:** The authors declare no conflict of interest in relation to the issues discussed in this manuscript.

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
