# Peer review of "Yam Staking Reduces Soil Loss Due to Crop Harvesting under Agronomic Management System: Environmental Effect of Soil Carbon Loss"

_agronomy, doi:10.3390/agronomy12123024_

Round 1
Reviewer 1 Report
The study investigated the effect of yam staking on soil loss due to crop harvesting. The study is interesting for me. Previous agronomic practice mainly focused on crop yield, and associating agronomic practice with soil carbon loss and carbon sequestration provided a new perspective to reduce carbon efflux. The results showed that staking reduced soil loss due to crop harvesting and increased yam yield. The reducing soil and nutrient loss under staking treatment was mainly caused by the reduced soil moisture and root hairs. The experiment and results have been detailed introduced in the paper, and could support the author's viewpoints. I still have some comments and suggestions.
First, the study was carried out for three years, and the results were showed for each year, separately. Is three any different among the three years? Whether the precipitation affect soil loss and carbon loss.
Second, please give the n value for each analysis, also the error bar in the figure (figure 5).
Third, does root hair mean small root on the tuber?? In my memory, root hairs are cylindrical extensions of root epidermal cells. Can “root hair” be used here?
Forth, what does the soil P, K, Ca, Mg loss mean? Soil total P, K, Ca, Mg loss or soil available P, K, Ca, Mg loss? According to the method in Line 133-137, these seems like available nutrient. Am I right? Please make it clear in the figure and also in the text.
Fifth, in this study, the paper focused on carbon loss mainly through soil loss due to crop harvesting. When considering total soil carbon pool and soil carbon sequestration, the change of soil carbon content was also important. Would the staking practice affect soil carbon content??
Sixth, I can’t follow the last sentence in the abstract line34. “Thus, yam staking mitigate soil loss and its carbon loss which can decrease sequestration potential of soil carbon stock.” If yam staking can mitigate soil loss and its carbon loss, the soil carbon sequestration potential may increase??
Line176-178: what’s the purpose to assess the mound dimensions (slant height, perpendicular height, base circumference and volume) at the end of every growing season?
Author Response
A point-by-point response to the comments of Reviewer 1
Thank you very much for giving us the opportunity to revise our manuscript entitled “Yam staking reduces soil loss due to crop harvesting under agronomic management system: Environmental effect of soil carbon loss” (agronomy-1844463). We have now revised the manuscript according to the comments of reviewer 1. We appreciate the reviewer for his/her suggestions which have greatly improved the manuscript. We have pasted reviewer comments below and provided our responses point by point.
We look forward to your positive consideration of our manuscript.
Thank you.
Yours sincerely,
Associate Professor Hanqing Yu
Corresponding author

Reviewer 2 Report
The manuscript concerns the effect of yam crops agronomic management on soil loss due to harvesting. The topic is of current interest and the data shown add meaningful informations. The scientific quality of the work is good, but something should be improved in the text. The description of methods must be more accurate, using a scientific language, citing opportunely the references to which each method is referred, and describing materials and instruments (including models and suppliers of instruments and reagents). The description of soil sampling anaysis (Section 2.2) must include all the sampling dates, the number of samples, repetitions, how the samples were collected, transported, stored, prepared. The pictures of Figure 3 are of very bad quality. A non-existent table 4 is frequently cited in the text. The Section 3.3 is missing. All the Figures and Tables captions must be completed with full description of abbreviations, symbols etc in order to improve their readability, and checked for some mistake. Several spelling and typo mistakes are also present throughout the whole manuscript.
Other comments:
Line 25: Dioscorea rotundata: italics.
Lines 30-31, 45, 112-113 and so on: check where lowercase and uppercase or superscript notations must be corrected.
Line 88: population: what is meant? aboundance, morphology or some other? please specify.
Section 2.1 site description: please specify if field is flat.
Line 117: how many samples? how many repetitions? how statistical analysis was carried out? How the samples were transported and stored until analysis?
Lines 133-134: Please rephrase the sentence citing the Olsen method to determine the plant-available phosphorus.
Line 135: “exchangeable bases”: Does it means the soil cation exchange capacity? Explain and describe the analysis method more in detail.
Lines 136-138: Please cite the reference to published methods adopted.
Lines 154-162: There is something confusing; there is no equation 4 shown; what is and how was determined the gravimetric moisture content? Please rephrase the sentences and check mistakes.
Lines 245-264: This section should be moved to Discussion.
Line 340: RHD or RHW? HGMDR or HGMDM?
Line 353: Where is table 4? Is it Table 3? The cited r value in table 3 is 0.92.
Lines 409 and 416: There is no Table 4. Maybe is Table 3.
Lines 415-416: Which are the data supporting this statement?
Lines 437-438: The meaning of this sentence is not very clear.
Author Response
A point-by-point response to the comments of Reviewer 2
Thank you very much for giving us the opportunity to revise our manuscript entitled “Yam staking reduces soil loss due to crop harvesting under agronomic management system: Environmental effect of soil carbon loss” (agronomy-1844463). We have now revised the manuscript according to the comments of reviewer 2. We appreciate the reviewers for their suggestions which have greatly improved the manuscript. We have pasted reviewer comments below and provided our responses point by point.
We look forward to your positive consideration of our manuscript.
Thank you.
Yours sincerely,
Associate Professor Hanqing Yu
Corresponding author
